# Mpox Outbreak 2022: A Comparative Analysis of the Characteristics of Individuals Receiving MVA-BN Vaccination and People Diagnosed with Mpox Infection in Milan, Italy

**DOI:** 10.3390/pathogens12091079

**Published:** 2023-08-24

**Authors:** Flavia Passini, Angelo Roberto Raccagni, Sara Diotallevi, Riccardo Lolatto, Elena Bruzzesi, Caterina Candela, Costanza Bertoni, Benedetta Trentacapilli, Maria Francesca Lucente, Antonella Castagna, Silvia Nozza

**Affiliations:** 1Infectious Diseases Unit, Vita-Salute San Raffaele University, 20132 Milan, Italy; f.passini@studenti.unisr.it (F.P.); bruzzesi.elena@hsr.it (E.B.); candela.caterina@hsr.it (C.C.); bertoni.costanza@hsr.it (C.B.); trentacapilli.benedetta@hsr.it (B.T.); lucente.francesca@hsr.it (M.F.L.); castagna.antonella1@hsr.it (A.C.); nozza.silvia@hsr.it (S.N.); 2Infectious Diseases Unit, San Raffaele Scientific Institute, 20132 Milan, Italy; diotallevi.sara@hsr.it (S.D.); lolatto.riccardo@hsr.it (R.L.)

**Keywords:** mpox, monkeypox, MVA-BN, vaccination, sexual behaviors, chemsex, key populations, men who have sex with men

## Abstract

Mpox caused a worldwide outbreak in 2022, disproportionately affecting MSM reporting high-risk sexual behaviors. The aim of this study was to compare the characteristics of people receiving MVA-BN vaccination with those of individuals diagnosed with mpox to guide future vaccination policies. This was a retrospective study on people with mpox infection or vaccination at San Raffaele Scientific Institute, Milan, Italy, from May to November 2022. Characteristics were compared using Mann–Whitney or chi-square/Fisher’s exact tests; multivariable logistic regression and classification tree analysis were applied. Overall, 473 vaccinated individuals and 135 with mpox were included; 472/473 and 134/135 were MSM. People with mpox were more frequently living with HIV (48.9% vs. 22.4%, *p* < 0.001), had ≥1 previous STI (75.6% vs. 35.7%, *p* < 0.001), were chemsex users (37.8% vs. 6.34%, *p* < 0.001), were with a higher number of partners (23.0% vs. 1.69%, *p* < 0.001), and had engaged in group sex (55.6% vs. 24.1%, *p* < 0.001). At multivariable analysis, PLWH (aOR = 2.86, 95%CI = 1.59–5.19, *p* < 0.001), chemsex users (aOR = 2.96, 95%CI = 1.52–5.79, *p* = 0.001), those with previous syphilis (aOR = 4.11, 95%CI = 2.22–7.72, *p* < 0.001), and those with >10 partners (aOR = 11.56, 95%CI = 6.60–21.09, *p* < 0.001) had a higher risk of infection. This study underscores the importance of prioritizing MSM with prior STIs and multiple partners as well as chemsex users in vaccination policies to curb mpox spread. A destigmatized assessment of sexual history is vital for comprehensive sexual health strategies.

## 1. Introduction

Mpox (formerly known as “monkeypox”) is a zoonotic disease caused by the mpox virus (MPXV) [1]. Human mpox caused an outbreak throughout Europe and North America that started in May 2022, and in July, the World Health Organization (WHO) proclaimed it a public health emergency of international concern (PHEIC) [2,3]. By May 2023, there had been a total of 87.314 confirmed cases of mpox, resulting in 129 deaths across 111 countries [2,3]. The Centers for Disease Control and Prevention (CDC) reported that the majority of those infected were young or middle-aged men [4]. Sexual activity has been identified as the primary risk factor for infection since the onset of the current outbreak [5,6,7]. Many initial cases of mpox were linked to participants in the International Pride Parade on the Spanish island of Gran Canaria, leading to a subsequent spread across various European countries [5]. Some individuals disclosed attending large gatherings or social events, having multiple sexual partners in the preceding weeks, and engaging in sexual activity while using recreational drugs [5,7]. The outbreak has disproportionately affected men who have sex with other men (MSM) and people living with HIV (PLWH), with locally acquired community transmission predominating in all countries during the summer of 2022 [5,6,7,8]. Italy recorded 957 confirmed mpox cases as of May 2023, with the Lombardy region having the highest number of cases, followed by Lazio and Emilia-Romagna [9]. While historical transmission occurred through direct contact with infected animals, in the current outbreak, human-to-human transmission is the predominant way of infection [10]. This is primarily facilitated by intimate, frequent skin-to-skin contact [10,11,12,13]. Direct contact with mpox rashes, scabs, oral and upper respiratory secretions, as well as genital and anal regions increases the risk of infection [10,11,12,13]. Additionally, asymptomatic mpox cases have been reported, raising concerns about potential asymptomatic viral shedding [14,15,16]. The transmission of mpox through clothing, towels, fetish gear, or sex toys is expected to be lower [10,11,12,13]. Notably, mpox has also been detected in the seminal fluids of infected individuals, suggesting that sexual activity may contribute to the spread [17]. The occurrence of concomitant sexually transmitted infections (STIs) has also been frequently observed, corroborating the key role of sexual transmission [5,7,8,18]. To prevent mpox infection, the CDC has recommended several measures [19]. Individuals diagnosed with mpox are required to isolate at home until clinically healed [19,20]. Safe sex practices and avoiding social gatherings have also been paramount during the current outbreak [19,20]. Authorities recommended limiting the number of sexual partners and avoiding high-risk spaces like dark rooms, saunas, sex clubs, and private or public sex parties [19,20]. Sharing sex toys, fetish gear, or towels among individuals with risk factors for infection should also be avoided [19,20]. Vaccination was an important tool in preventing the spread of mpox during the 2022 outbreak [21]. The JYNNEOS^®^ (Bavarian Nordic, Hellerup, Denmark) vaccine (also known as Modified Vaccinia Ankara Bavarian Nordic (MVA-BN)) is used to prevent both smallpox and mpox among people at high risk for infection. MVA-BN is a third-generation vaccine based on a live, attenuated Orthopoxvirus, which does not replicate efficiently in humans [21]. While initial findings indicate that the first MVA-BN dose offers some protection, two doses are currently recommended for stronger immunization [21,22].

During the 2022 outbreak, the CDC recommended MVA-BN immunization for MSM, transgender, or nonbinary individuals who reported high-risk sexual behaviors (based on the number of partners, previous STIs, sex work, or attendance at large sexual gatherings), those with known exposure to mpox virus, and those with potential occupational risks [21]. Italian guidelines for the 2022 mpox outbreak suggested using MVA-BN for high-risk individuals older than 18 years old [23]. Pre-exposure vaccination as prophylaxis was offered to healthcare personnel with possible direct exposure and members of key populations (e.g., gay, trans, bisexual, and MSM) engaging in high-risk sexual behaviors [23]. Despite international and local recommendations for mpox vaccination, all individuals face a potential risk of infection regardless of sex, sexual orientation, or lifestyle. Indeed, a large international case series described several mpox cases among women and nonbinary individuals [24]. Stigmatization has characterized the current mpox outbreak from its beginning, leading the WHO to release a statement declaring “mpox” a preferred term over monkeypox to avoid stigma [25]. However, in an outbreak scenario with limited available vaccines, prioritizing key groups and strongly encouraging the MVA-BN vaccine for individuals at substantial risk of acquiring mpox are crucial to mitigate transmission.

The primary objective of this study is to compare the characteristics of individuals who received MVA-BN vaccination with those diagnosed with mpox infection. This comparison aims to assess the efficiency of MVA-BN vaccination delivery during the 2022 mpox vaccination campaign, specifically in terms of coverage among people at substantial risk of mpox infection and its reach to all key populations, and to identify missed opportunities for prevention, having the characteristics of people diagnosed with mpox infection as a reference for this evaluation. Secondarily, mpox symptom presentation and the clinical course of the disease are described to grant further insight into the underlying characteristics which might explain the community transmission of the infection. Lastly, current mpox vaccination guidelines and risk factors considered in the recommendations for vaccination are compared with the demographical data, medical history, and sexual behaviors of people who have been diagnosed with mpox to facilitate a reflection on the optimal vaccine prioritization during current or future mpox outbreaks, ensuring that those at the highest risk are appropriately targeted.

## 2. Materials and Methods

This is a retrospective study involving people diagnosed with mpox infection between May and November 2022 and individuals who were vaccinated via the intradermal or subcutaneous route with MVA-BN between August and October 2022 at the Infectious Diseases Unit of IRCCS San Raffaele, Milan, Italy. 

### 2.1. Cases Definition

#### 2.1.1. Mpox-Infected Individuals

Mpox cases with PCR-confirmed infection on a swab were included. Cases with suspected mpox infection received a physical examination and were tested for mpox with PCR on oropharyngeal, anal, genital, and cutaneous swabs plus blood, urine, and seminal fluids. Approximately once per week, people repeated clinical evaluation and further research for mpox on the same samples, based upon medical judgment. Concurrent STIs, including *Chlamydia trachomatis* and *Neisseria gonorrhoeae*; HIV or HIV-RNA; syphilis; and viral hepatitis were tested. HIV-RNA and CD4^+^ lymphocytes were assessed at the time of mpox diagnosis among PLWH. The real-time polymerase chain reaction (RT-PCR) assay RealStar^®^ Orthopoxvirus PCR Kit 1.0 (altona DIAGNOSTICS—Milan, Italy), targeting *variola* virus and non-variola Orthopoxvirus species (*cowpox*, mpox, *raccoonpox*, *camelpox*, and *vaccinia virus*), was used to detect the presence of non-variola DNA. A specific RT-PCR targeting mpox virus DNA (Liferiver, Shanghai ZJ Bio-Tech Co., Shanghai, China) was subsequently used for mpox infection confirmation. Samples were considered positive with a Cycle Threshold (CT) value of RT-PCR ≤ 40. Laboratory analyses were performed at the Laboratory of Clinical Microbiology, Virology, and Bioemergencies of Luigi Sacco University Hospital, Milan, an Italian reference center for mpox diagnosis. On their first visit, individuals filled out a questionnaire on their previous clinical history and high-risk sexual behaviors referring to the 3 months before access to the clinic. 

#### 2.1.2. Mpox-Vaccinated Individuals

Individuals who received mpox vaccination with MVA-BN with available clinical history were considered. People scheduled the vaccination through the “Sistema Sanitario della Regione Lombardia” (SSR) website, which allowed people residing in Lombardy to independently schedule an appointment following the Italian Government’s recommendations for mpox vaccination. The vaccination campaign ran from 9 August to 31 October 2022, and around 1800 people received shots overall. Medical counseling on HIV, STI prevention, HIV pre-exposure prophylaxis (PrEP), and mpox transmission pathways was provided before the vaccination was administered. All people were given the option of using our walk-in service to receive testing for HIV and STIs and to receive PrEP. Every vaccinee had the option of voluntarily completing a questionnaire to verify if they fit the criteria for mpox vaccination of the local health authority (ATS Lombardia and Ministry of Health). The survey was designed to evaluate people’s sexual behavior that could put them at risk for mpox infection. The questionnaire included seven questions with multiple choices, and it was self-completed. The first question investigated the number of sexual partners in the previous three months; the possible answers were one, two to ten, eleven to thirty, thirty-one to fifty, and more than fifty. The second question considered participation in group sex, while the third considered participation in sexual encounters in clubs or cruising activities. Cruising was defined as walking or driving about certain areas (e.g., train and gas stations, woods, and public places), called cruising grounds, looking for a sexual partner. The fourth question asked about previous STIs in the last year. The fifth regarded the use of chemsex. Chemsex was defined as the use of either cocaine, methamphetamine (stimulant), mephedrone, gamma hydroxybutyrate (GHB) and gamma-butyrolactone (GBL), and methylenedioxypyrovalerone (MDPV) during or before sex. 

### 2.2. Statistical Analysis

The median (1st quartile, 3rd quartile) and the frequency (%) were used, as appropriate, to describe the individuals’ characteristics. A Mann–Whitney rank-sum test and chi-square test were used to compare the characteristics of people vaccinated with MVA-BN and diagnosed with mpox for continuous and categorical variables, respectively. Multivariable logistic regressions and classification tree analysis were applied. Variables with a probability value (*p*-value) < 0.1 in the univariable analyses were entered in the multivariable multinomial logistic regression model. A two-sided probability value (*p*-value) < 0.05 was considered statistically significant. Analyses were carried out using R Statistical Software, version 4.2.2 (R Foundation for Statistical Computing, Vienna, Austria). Undetectable HIV-RNA < 50 copies/mL was defined as “undetectable”, detectable HIV-RNA < 50 copies/mL as “residual” viremia, and detectable HIV-RNA > 50 copies/mL as “positive” viremia. The information of interest for this study was collected from the individuals followed as part of routine clinical care and recorded in the database of the Infectious Diseases Unit of the San Raffaele Scientific Institute (CSL Cohort). On their first visit, individuals provided written informed consent on the use of their data in scientific analyses. The data used in the analyses of this study were extracted from the CSL Cohort database on the 10 May 2023 (freezing date).

## 3. Results

Overall, 608 individuals were included, of which 135 were infected with mpox and 473 were vaccinated with MVA-BN. 

### 3.1. People’s Characteristics

#### 3.1.1. Mpox-Infected

Of the 135 mpox cases, the median age was 37.9 (interquartile (IQR): 33.6–43.3) and 134 (99.3%) were male, all of whom were MSM (100%). Among the mpox-infected, 66 (48.9%) individuals were living with HIV (PLWH) and 48 individuals were PrEP users (35.6%). Among PLWH, the median years from HIV diagnosis were 9.44 (IQR: 6.23–14.8). The median level of the nadir of CD4^+^ lymphocyte count was 440 cells/μL (IQR: 248–574). At the time of mpox diagnosis, the median CD4^+^ count was 695 cells/μL (IQR: 588–908), the median CD4% was 35.3% (IQR: 29.4–39.3%), and the median CD4^+^/CD8^+^ ratio was 0.98 (IQR: 0.62–1.21). Regarding PLWH, 6 (9.84%) had positive HIV-RNA, 34 (55.7%) had undetectable HIV-RNA, and 21 (34.4%) had residual HIV-RNA. There were five people (3.70%) who had a previous AIDS-defining event. A history of HCV infection was reported in 10 people (7.41%) (data were missing for 20 (14.8%)), whereas a history of HBV was reported in 1 case (0.74%) (data were missing for 70 (51.9%)). Previous STIs were detected in 102 people (75.6%), of which 55 (40.7%) had chlamydia, 60 (44.4%) had gonorrhea, 74 (54.8%) had syphilis, 6 (4.44%) had sexually transmitted enteric infections, and 36 (26.7%) had other forms of less common STIs (e.g., mycoplasma and ureaplasma). Overall, 51 individuals (37.8%) were chemsex users and 75 (55.6%) participated in group sex. According to the self-disclosed number of sexual partners in the previous three months, 2 people (1.48%) declared they had had sex with just 1 partner, 23 (17.0%) with 2 to 10 partners, 68 (50.4%) with 11 to 30 partners, 11 (8.15%) with 31 to 50 partners, and 31 (23.0%) with more than 50 partners. In detail, 83 individuals (61.4%) specified the type of sexual activity: 62 (74.7%) had receptive anal sex, 67 (80.7%) had insertive anal sex, 76 (91.6%) performed oral sex, and 74 (89.2%) received oral sex. Overall, 73 individuals (54.1%) disclosed engaging in cruising activities and 46 cases (34.1%) came in close contact with another known mpox case. The complete characteristics of mpox-infected individuals are presented in Table 1.

#### 3.1.2. Vaccinated

Of the 473 vaccinated individuals, the median age was 36.4 (IQR: 31.4–43.3) and all were male (*n* = 473, 100%), of which 472 (99.8%) were MSM. Overall, 106 (22.4%) were PLWH, the median years since HIV diagnosis were 10.1 (IQR: 6.60–16.7), and 152 individuals were PrEP users (32.1%). Among PLWH, the median nadir of the CD4^+^ count was 384 cells/μL (IQR: 220–619). At the time of MVA-BN vaccination, the median CD4^+^ count was 744 cells/μL (IQR: 590–859), the median CD4% was 33% (IQR: 29.4–39.3%), and the median CD4^+^/CD8^+^ ratio was 0.88 (IQR: 0.72–1.13). Regarding PLWH, 3 (3.80%) had positive HIV-RNA, 46 (58.2%) had undetectable HIV-RNA, and 30 (38.0%) had residual HIV-RNA. There were four people (0.85%) with a previous AIDS-defining event. A history of HCV infection was reported in 17 people (3.59%) (data were missing for 294 (62.2%)), whereas a history of HBV was reported in 3 cases (0.63%) (data were missing for 372 (78.6%)). Previous STIs were seen in 169 cases (35.7%), of which 41 (8.67%) had chlamydia, 43 (9.09%) had gonorrhea, 46 (9.73%) had syphilis, 5 (1.06%) had sexually transmitted enteric infections, and 28 (5.92%) had other less common forms of STIs. Overall, 30 individuals (6.34%) were chemsex users and 114 (24.1%) participated in group sex. According to the self-disclosed number of sexual partners in the previous three months, 80 people (16.9%) declared they had had sex with just 1 partner, 283 (59.8%) with 2 to 10 partners, 87 (18.4%) with 11 to 30 partners, 15 (3.17%) with 31 to 50 partners, and 8 (1.69%) with more than 50 partners. Lastly, 133 individuals (28.1%) disclosed engaging in cruising activities. The complete characteristics of mpox-vaccinated individuals are presented in Table 1.

### 3.2. Comparison of the Characteristics between Mpox-Infected and -Vaccinated Individuals

#### 3.2.1. Univariable Analysis

When comparing the characteristics of individuals vaccinated with MVA-BN and those with mpox infection, people diagnosed with mpox were more frequently living with HIV than vaccinated ones (48.9% (n = 66) versus 22.4% (n = 106), *p* < 0.001), and more had infection with HCV (7.41% (n = 10) versus 3.59% (n = 17), *p* < 0.001) and HBV (0.74% (n = 1) versus 0.63% (n = 3), *p* < 0.001). Previous STIs were more common among the mpox-infected group (75.6% (n = 102) versus 35.7% (n = 169), *p* > 0.001). In particular, chlamydia (40.7% (n = 55) versus 8.67% (n = 41), *p* > 0.001), gonorrhea (44.4% (n = 60) versus 9.09% (n = 43), *p* < 0.001), syphilis (54.8% (n = 74) versus 9.73% (n = 46), *p* < 0.001), sexually transmitted enteric infections (4.44% (n = 6) versus 1.06% (n = 5), *p* < 0.001), and other forms of STIs (26.7% (n = 36) versus 6.34% (n = 30), *p* < 0.001) were more common among the mpox-infected. Individuals with an mpox diagnosis were more commonly chemsex users (37.8% (n = 51) versus 6.34% (n = 30), *p* < 0.001) and declared engaging in group sex more (55.6% (n = 75) versus 24.1% (n = 114), *p* < 0.001). 

The number of partners in the previous three months (1 partner: 1.48% (n = 2) versus 16.9% (n = 89); 2–10 partners: 17.0% (n = 23) versus 59.8% (n = 283); 11–30 partners: 50.4% (n = 68) versus 18.4% (n = 87); 31–50 partners: 8.15% (n = 11) versus 3.17% (n = 15+); >50 partners: 23.0% (n = 31) versus 1.69% (n = 8); *p* < 0.001) and engagement in cruising (54.1% (n = 73) versus 28.1% (n = 133), *p* < 0.001) were higher in the infected group. No differences were identified regarding PrEP use (35.6% (n = 48) versus 32.1% (n = 152), *p* = 0.521), median age (37.9 years (IQR: 33.6–43.3) versus 36.4 years (IQR: 31.4–43.3), *p* = 0.072), years from HIV diagnosis (9.44 (IQR: 6.23–14.8) versus 10.1 (IQR: 6.60–16.7), *p* = 0.864), nadir CD4^+^ count (440 (IQR: 248–574) versus 384 (IQR: 220–619), *p* = 0.897), CD4^+^ count (695 (IQR: 588–908) versus 744 (IQR: 590–859), *p* = 0.995), CD4% (35.3 (IQR: 29.4–39.3) versus 33.0 (IQR: 28.4–38.4), *p* = 0.688), and CD4^+^/CD8^+^ ratio (0.98 (IQR: 0.62–1.21) versus 0.88 (IQR: 0.72–1.13), *p* = 0.752) at infection or vaccination. The complete characteristics of the included individuals are presented in Table 1. 

#### 3.2.2. Multivariate Analysis

At multivariable analysis, the risk of mpox infection compared to MVA-BN vaccination was higher among PLWH (adjusted odds ratio (aOR) = 2.86, 95% CI = 1.59–5.19, *p* < 0.001), chemsex users (aOR = 2.96, 95% CI = 1.52–5.79, *p* = 0.001), and those with previous syphilis (aOR = 4.11, 95% CI = 2.22–7.72, *p* < 0.001) or with more than ten partners in the previous three months (aOR = 11.56, 95% CI = 6.60–21.09, *p* < 0.001). The multivariate analysis is presented in Table 2. 

#### 3.2.3. Classification Tree Analysis

Classification tree analysis showed that, among people with more than ten partners and with previous syphilis (74, 12.17%), 64 out of 74 (86.4%) were diagnosed with mpox. Among those with more than 10 partners, with no previous syphilis, and with previous gonorrhea (28, 4.61%), 19 out of 28 (67.8%) had an mpox diagnosis. Considering individuals without previous syphilis or gonorrheal infection (118, 19.41%), 91 out of 118 (77.1%) were vaccinated against mpox. Lastly, among individuals with ten or fewer partners (388, 63.82%), 363 out of 388 (93.5%) received mpox vaccination (Figure 1).

## 4. Discussion

The traits that might have increased the likelihood of mpox infection and the present immunization practices were examined in this retrospective analysis. Data gathered during medical visits and the performed statistical analysis supported the intention to urge key populations to receive the mpox immunization and allow them to be prioritized in vaccination strategies; this was implemented from the very beginning of the vaccination campaign, following early data from international cohorts. According to the results obtained, mpox-infected individuals were almost entirely male, all of whom were MSM. Multiple research projects carried out in numerous countries with various sample populations have shown that MSM have always been emphasized as the major key population involved in the current outbreak. It has been demonstrated that intimate physical contact during sexual activity is heavily involved in the transmission of mpox [26,27]. We analyzed the prior STIs and sexual behavior of mpox-infected individuals to determine which were the most common risk factors. Being close to other mpox individuals is unquestionably the main risk factor for contracting the virus. The majority of the participants in our study group previously had an STI, including syphilis, gonorrhea, and chlamydia, supporting the results of many other studies [5,7,8,18,28,29], indicating a risky sexual life. More than half of the group who contracted mpox disclosed engaging in group sexual activity. Group intercourse, of course, increases a person’s risk of contracting not only mpox but also all other STIs, including HIV. Chemsex use was another trait shared by those with an mpox diagnosis. People that engage in chemsex put themselves at risk since they can forget to wear condoms or take PrEP. A factor that we demonstrated to be crucial was the self-disclosed number of sexual partners in the previous three months. Having sex with numerous partners significantly raises the likelihood of getting mpox, whereas only having sex with one partner significantly lowers that risk. More than half of those who had the mpox virus also participated in cruising. PLWH with mpox diagnoses raised the alarm worldwide until the CDC established guidelines for the prevention and treatment of mpox in patients with HIV infection [30]. Our data support this hypothesis, as nearly half of the infected were PLWH, while fewer than half of them utilized PrEP, suggesting an existing barrier for HIV prevention [31].

As previously indicated, mpox disproportionately afflicted MSM; nevertheless, framing the mpox outbreak as entirely or primarily occurring among MSM and spreading through sexual activity may increase stigma, recalling what happened during the HIV/AIDS epidemic in the 1980s. Anyone can spread mpox, regardless of sexual preference. Because false information can travel rapidly and easily in the digital era, this risk is extremely high [32]. Since mpox, as seen in the clinical characteristics section, can spread in different ways, not only sexually, it is crucial to realize that not just MSM, men, or PLWH are at risk. However, vaccination policies must be prioritized to protect the key populations at higher risk. Numerous studies have shown that the MVA-BN vaccination reduces the chances of being infected with mpox [22]. Nevertheless, further precautions to avoid exposure should be taken, especially among immunocompromised individuals. Regardless of the delivery route, both the first and second doses offer significant protection against mpox [22]. To determine the length of protection, which may vary depending on the number of doses or the mode of administration, additional research needs to be conducted. MVA-BN vaccination coverage among persons at risk is low, and many eligible persons have not received both doses. For optimal protection, persons at risk for mpox should receive the two-dose series, as recommended by vaccination guidelines [21]. According to the guidelines, the vaccinated individuals analyzed in our study were almost entirely MSM and two-thirds of them had a previous STI in the last year. In fact, the CDC’s guidelines suggest vaccination with MVA-BN for people who are “gay, bisexual, or MSM or a transgender, nonbinary, or gender-diverse person who in the past 6 months has had or a new diagnosis of one or more sexually transmitted diseases (e.g., chlamydia, gonorrhea, or syphilis) or more than one sex partner” [21]. It should be emphasized that, as previously demonstrated, women can also contract mpox infection, albeit to a lesser extent [24]. Our study does not aim to define new guidelines for the vaccination campaign, but rather to clarify the risk factors that should be considered when prioritizing the vaccination campaign. Risk factors can be present in anyone, not necessarily among MSM; therefore, access to the vaccine should never be hindered, but rather just prioritized. Our purpose is to raise awareness about the importance of ensuring access to vaccination, considering the positivity to mpox even in women and in the categories we defined as not at risk. Furthermore, according to our results, the number of sexual partners in the last three months statistically increased the likelihood of becoming infected. Also, vaccination was common among those who had any sexual intercourse with strangers, as suggested by vaccination policies. No mention was made in the CDC’s vaccination policies about chemsex users [21]. However, according to our results, chemsex users are statistically higher among infected individuals. The use of chemsex significantly increases the risk of transmission for HIV and other STIs, but probably also for mpox infection. Generally, according to our data, we can state that the vaccination guidelines were followed. 

Our study was subject to some limitations. The first limitation concerned missing data, as there may have been some data that were not reported by the participants. Furthermore, participation was voluntary; any individual could choose not to fill out the questionnaire. Another challenge of the study was pinpointing exact sexual behavior, as most participants were fearful of intimate information and did not always openly disclose their sexual behavior even if self-reported, with possible social desirability bias. Lastly, despite the extensive data we obtained, the study was conducted only among people that were referred to our center for mpox infection or mpox vaccination, which might limit the generalization of our data. However, the risk traits described here are very similar to what were observed in other international cohorts.

## 5. Conclusions

This retrospective study validates previous studies reporting how the mpox outbreak of 2022 has affected the world as a whole. Even though mpox can be transmitted during sexual activity through close contact, it should not be merely classified as a conventional STI. It is essential to share culturally appropriate health promotion messages and to deliver clear and timely information on the signs and symptoms of mpox and how to prevent it. Since those with an mpox diagnosis more frequently exhibited risky sexual behaviors than those who received the MVA-BN vaccine, it is important that vaccination policies give priority to the key populations in order to prevent the spread of the virus, and a comprehensive assessment of sexual history, without stigma, is paramount to addressing sexual health needs.

## Figures and Tables

**Figure 1 pathogens-12-01079-f001:**
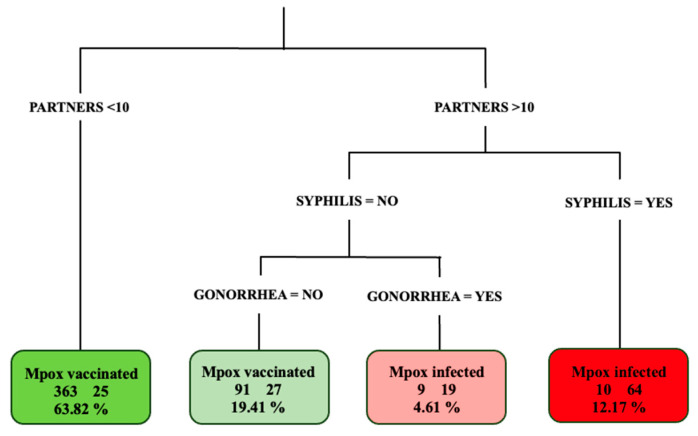
Classification tree analysis of the characteristics of included individuals according to mpox infection or vaccination status.

**Table 1 pathogens-12-01079-t001:** Characteristics of the included individuals according to mpox infection or vaccination status.

Charcateristics	Overall (N = 608)	Mpox-Vaccinated (N = 473)	Mpox-Infected (N = 135)	*p*-Value
Age	36.7 [31.7; 43.4]	36.4 [31.4; 43.3]	37.9 [33.6; 43.3]	0.072
Male gender	607 (99.8%)	473 (100%)	134 (99.3%)	0.222
MSM	607 (99.8%)	472 (99.8%)	135 (100%)	1.000
Living with HIV	172 (28.3%)	106 (22.4%)	66 (48.9%)	**<0.001**
HIV-RNA * ^:				0.366
Undetectable	80 (57.1%)	46 (58.2%)	34 (55.7%)	
Residual	51 (36.4%)	30 (38.0%)	21 (34.4%)	
Positive	9 (6.43%)	3 (3.80%)	6 (9.84%)	
Years from HIV diagnosis *	9.78 [6.38; 15.5]	10.1 [6.60; 16.7]	9.44 [6.23; 14.8]	0.864
Years of ART *	0.00 [0.00; 2.94]	0.00 [0.00; 0.00]	0.00 [0.00; 7.53]	<0.001
Nadir of CD4^+^ *	418 [223; 583]	384 [220; 619]	440 [248; 574]	0.897
AIDS *	9 (1.48%)	4 (0.85%)	5 (3.70%)	0.029
CD4^+^ count *	715 [590; 898]	744 [590; 859]	695 [588; 908]	0.995
CD4^+^% count *	34.2 [28.5; 38.7]	33.0 [28.4; 38.4]	35.3 [29.4; 39.3]	0.688
CD4^+^/CD8^+^ *	0.89 [0.66; 1.19]	0.88 [0.72; 1.13]	0.98 [0.62; 1.21]	0.752
PrEP users	200 (32.9%)	152 (32.1%)	48 (35.6%)	0.521
HCV	27 (4.44%)	17 (3.59%)	10 (7.41%)	<0.001
Unknown	314 (51.6%)	294 (62.2%)	20 (14.8%)	
HBV	4 (0.66%)	3 (0.63%)	1 (0.74%)	**<0.001**
Unknown	442 (72.7%)	372 (78.6%)	70 (51.9%)	
Previous STIs	271 (44.6%)	169 (35.7%)	102 (75.6%)	**<0.001**
Chlamydia	96 (15.8%)	41 (8.67%)	55 (40.7%)	**<0.001**
Gonorrhea	103 (16.9%)	43 (9.09%)	60 (44.4%)	**<0.001**
Syphilis	120 (19.7%)	46 (9.73%)	74 (54.8%)	**<0.001**
STEIs	11 (1.81%)	5 (1.06%)	6 (4.44%)	0.019
Other STIs	64 (10.5%)	28 (5.92%)	36 (26.7%)	**<0.001**
Chemsex users	81 (13.3%)	30 (6.34%)	51 (37.8%)	**<0.001**
Number of partners				<0.001
1	82 (13.5%)	80 (16.9%)	2 (1.48%)	
2–10	306 (50.3%)	283 (59.8%)	23 (17.0%)	
11–30	155 (25.5%)	87 (18.4%)	68 (50.4%)	
31–50	26 (4.28%)	15 (3.17%)	11 (8.15%)	
>50	39 (6.41%)	8 (1.69%)	31 (23.0%)	
Engaged in group sex	189 (31.1%)	114 (24.1%)	75 (55.6%)	**<0.001**
Cruising	206 (33.9%)	133 (28.1%)	73 (54.1%)	**<0.001**

* Referring to PLWH. ^ Undetectable: undetectable HIV-RNA < 50 copies/mL; Residual: detectable HIV-RNA < 50 copies/mL; Positive: detectable HIV-RNA > 50 copies/mL. Note. Bold: significant *p*-values.

**Table 2 pathogens-12-01079-t002:** Multivariate analysis of the risk of mpox infection according to individuals’ characteristics and risk factors.

Characteristics	Levels	Odds Ratio (95% CI)	*p*-Value
Living with HIV	Yes vs. No	2.86 (1.59; 5.19)	**<0.001**
Chlamydia	Yes vs. No	2.02 (0.96; 4.21)	0.06
Gonorrhea	Yes vs. No	2.00 (0.96; 4.14)	0.06
Syphilis	Yes vs. No	4.11 (2.22; 7.72)	**<0.001**
STEIs	Yes vs. No	1.45 (0.24; 10.16)	0.69
Chemsex	Yes vs. No	2.96 (1.52; 5.79)	**0.001**
Partners	(2, 3, 4) vs. (0, 1)	11.56 (6.60; 21.09)	**<0.001**

Note. Bold: significant *p*-values.

## Data Availability

Data supporting this study are available upon reasonable request from the corresponding author; the data are not publicly available due to ethical reasons.

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
