# Peer review of "Mpox Outbreak 2022: A Comparative Analysis of the Characteristics of Individuals Receiving MVA-BN Vaccination and People Diagnosed with Mpox Infection in Milan, Italy"

_pathogens, 2023, doi:10.3390/pathogens12091079_

Round 1

Reviewer 1 Report

Overall view of publication

The manuscript “Mpox Outbreak 2022: Comparative Analysis of the Individuals’ Characteristics Receiving MVA-BN Vaccination and People Diagnosed with Mpox Infection in Milan, Italy” described the findings of a retrospective study analysing the individual characteristics of MVA-BN vaccinated versus mpox infected individuals. The manuscript provided some important insights into the populations affected by mpox outbreak. It also demonstrated important differences between people receiving the MVA-BN vaccine and those that presented at the clinic with mpox infections. Some sections of the manuscript were very well written however other sections require significant modifications.

I found that the discussion and conclusions sections were very well written and insightful. The discussion summarised the findings of this study well in the context of the bigger picture associated with the mpox outbreak. The authors also provided a good discussion of the limitations of this study, and they wrote a good conclusion.

By contrast, I found the introduction section very problematic. It was too long and had a lot of information that was irrelevant for this study. In addition, I found sections that were copied and pasted from elsewhere. For example, lines 129-141 were copied out the introduction of a previous publication from the group (Candela et al. (2023) doi: 10.3390/v15030667). There were also other sections, where it appeared as though the authors had copied information from a medical textbook or public health webste. Examples include:

           Measures for reducing risk of becoming infected with mpox (lines 106-110)

           The CDC guidelines for MVA-BN vaccination (lines 142-149)

           The description of mpox disease course (lines 72-93)

I also found that in the introduction, the authors were missing citations. As many statements were made without a reference. 

Similarly, to the introduction, I found that the abstract problematic. Although it summarises the aims, methods, and results, it appears as though sentences were copied from these sections and inserted into the abstract. It also lacked structure, a conclusion statement, and an introductory sentence about the mpox outbreak. I strongly recommend that the authors rewrite both the introduction and abstract. In the introduction, the authors need to focus on the background information most relevant for the study.

The methods and results sections were generally well written and only require minor modifications. Would the authors please made the following corrections.

Methods

            Line 216: Change “suggestions” to “criteria”.

            Line 218: Change “pose” to “put”.

Results

           The authors provided a very detailed description of the characteristics of the study participants, which was good.

           Line 246: Change “of which 135 infected” to “of which 135 were infected”

           Line 256: The median CD4% is stated in this line, but the data is missing from Table 1. Please the data in this table.

           Line 258: Change “had detectable HIV” to “had residual HIV”, so that it matches what is stated in Table 1.

           Line 259:  Change "microL" to "μL"

           To assist the reader in understanding the differences between the categories within the HIV-RNA variable, the threshold level of HIV-RNA for undetectable, residual and positive could be stated in Table 1. 

           Line 260: Add full stop after “event”.

           Line 280: Refer to the comment on line 256

           Line 283: Refer to the comment on line 259

           Line 286: Change “of which 41 (8.67%) chlamydia” to “of which 41 (8.67%) had chlamydia”.

Lines 320-321: The authors refer to Table 1, which summarises characteristics of the study participants. This information was summarised earlier in the results section (3.1.1 and 3.1.2). The table should be referenced in these sections also.

The quality of the English was overall good and only small modifications are needed, which are stated above.

Author Response

Reviewer 1 

Overall view of publication

The manuscript “Mpox Outbreak 2022: Comparative Analysis of the Individuals’ Characteristics Receiving MVA-BN Vaccination and People Diagnosed with Mpox Infection in Milan, Italy” described the findings of a retrospective study analysing the individual characteristics of MVA-BN vaccinated versus mpox infected individuals. The manuscript provided some important insights into the populations affected by mpox outbreak. It also demonstrated important differences between people receiving the MVA-BN vaccine and those that presented at the clinic with mpox infections. Some sections of the manuscript were very well written however other sections require significant modifications.

I found that the discussion and conclusions sections were very well written and insightful. The discussion summarised the findings of this study well in the context of the bigger picture associated with the mpox outbreak. The authors also provided a good discussion of the limitations of this study, and they wrote a good conclusion.

We thank the Reviewer for the time to revise this manuscript and for the very important suggestions, which helped strengthening this our findings and message. Please find below detailed answer to Reviewer comments.

By contrast, I found the introduction section very problematic. It was too long and had a lot of information that was irrelevant for this study. In addition, I found sections that were copied and pasted from elsewhere. For example, lines 129-141 were copied out the introduction of a previous publication from the group (Candela et al. (2023) doi: 10.3390/v15030667). There were also other sections, where it appeared as though the authors had copied information from a medical textbook or public health website. Examples include:

  • Measures for reducing risk of becoming infected with mpox (lines 106-110)
  • The CDC guidelines for MVA-BN vaccination (lines 142-149)
  • The description of mpox disease course (lines 72-93)

We thank the Reviewer for this important comment and apologize for the redundancy of the Introduction section. Introduction section has now been completely revised and significantly shortened. The introduction section now includes a short general overview on mpox and a more focused description on information relevant to this specific study (e.g. sexual behaviours of people diagnosed with mpox, indications for MVA-BN vaccinations etc), which are more in line with the study aim and more useful for correct interpretation of the discussion.

I also found that in the introduction, the authors were missing citations. As many statements were made without a reference. 

Thank you for this comment, we now included further reference for every statement made in the revised form of the Introduction section.

Similarly, to the introduction, I found that the abstract problematic. Although it summarises the aims, methods, and results, it appears as though sentences were copied from these sections and inserted into the abstract. It also lacked structure, a conclusion statement, and an introductory sentence about the mpox outbreak. I strongly recommend that the authors rewrite both the introduction and abstract. In the introduction, the authors need to focus on the background information most relevant for the study.

According to Reviewer suggestion, we modified the Abstract.

The methods and results sections were generally well written and only require minor modifications. Would the authors please made the following corrections.

According to Reviewer suggestion, we modified the Methods and Results sections, as detailed below.

Methods

  • Line 216:Change “suggestions” to “criteria”. Modified according to Reviewer suggestion.
  • Line 218:Change “pose” to “put”. Modified according to Reviewer suggestion.

Results

  • The authors provided a very detailed description of the characteristics of the study participants, which was good.
  • Line 246:Change “of which 135 infected” to “of which 135 were infected”. Modified according to Reviewer suggestion.
  • Line 256: The median CD4% is stated in this line, but the data is missing from Table 1. Please the data in this table. Thank you for noting this point.Data on CD4% count among people diagnosed with mpox and vaccinated individuals has been added in Table 1.
  • Line 258:Change “had detectable HIV” to “had residual HIV”, so that it matches what is stated in Table 1. We agree with Reviewer suggestion.We modified the manuscript text (mpox infection and vaccinated individuals sections) to match the “positive”, “residual” and “undetectable” HIV-RNA definitions used in Table 1.In order to improve clarity, we also included in the Methods section these definitions.
  • Line 259:  Change "microL" to "μL". We modified this through the manuscript according to Reviewer suggestion.
  • To assist the reader in understanding the differences between the categories within the HIV-RNA variable, the threshold level of HIV-RNA for undetectable, residual and positive could be stated in Table 1. 

We agree with Reviewer suggestion and included these definitions in Table 1. Moreover, in order to improve clarity, we also included in the Methods section these definitions.

  • Line 260:Add full stop after “event”. Thank you for catching this typo.
  • Line 280:Refer to the comment on line 256. Thank you for noting this point.Data on CD4% count among people diagnosed with mpox and vaccinated individuals has been added in Table 1.
  • Line 283:Refer to the comment on line 259. We modified this through the manuscript according to Reviewer suggestion.
  • Line 286:Change “of which 41 (8.67%) chlamydia” to “of which 41 (8.67%) had chlamydia”. We modified this sentence according to Reviewer suggestion.

Lines 320-321: The authors refer to Table 1, which summarises characteristics of the study participants. This information was summarised earlier in the results section (3.1.1 and 3.1.2). The table should be referenced in these sections also. We included reference to Table 1 also in sections 3.1.1 an 3.1.2.

Reviewer 2 Report

Thank you for providing this manuscript. 

I agree with you that the mpox outbreak of 2022 is a major public health event, affecting primarily the high risk MSM population.

However, I cannot follow how you came from the introduction (which summarizes the 2022 outbreak and many of its characteristics) to your research question (what exactly is your research question?), and finally, how the methodology you used is applicable at all. In fact, your research question and the aims and objectives are not really defined. What do you want to demonstrate with the differences in characteristics between vaccinated and infected subjects? Is it a different risk behaviour? It is for me hard to understand why you choose exactly vaccinated versus infected for your analysis, as on the one hand, vaccination is a voluntary and intentional act, whereas infection is an adverse, unplanned incidence. The rationale why you are comparing those two groups and what you aimed to demonstrate should clearly be described in your manuscript.

If you want to write a summary of the 2022 mpox outbreak, then a lot of the elements of your current draft can be re-used. But that would be another article type, more of a review, less of a research article.

In case you want to keep it as a research article, there needs to be more clarity why the comparison of vaccinated versus infected individuals is appropriate, what you want to demonstrate with it, and how you selected and interpreted the parameters where you saw differences between the groups. Also an interpretation what those differences mean in the context of your research question should be provided.

Author Response

Review 2

Thank you for providing this manuscript.

I agree with you that the mpox outbreak of 2022 is a major public health event, affecting primarily the high risk MSM population.

We thank the Reviewer for the time to revise this manuscript and for the helpful suggestions which contributed to strengthen the study aims, findings and overall message. Please find below detailed answer to Reviewer comments.

However, I cannot follow how you came from the introduction (which summarizes the 2022 outbreak and many of its characteristics) to your research question (what exactly is your research question?), and finally, how the methodology you used is applicable at all. In fact, your research question and the aims and objectives are not really defined. What do you want to demonstrate with the differences in characteristics between vaccinated and infected subjects? Is it a different risk behaviour? It is for me hard to understand why you choose exactly vaccinated versus infected for your analysis, as on the one hand, vaccination is a voluntary and intentional act, whereas infection is an adverse, unplanned incidence. The rationale why you are comparing those two groups and what you aimed to demonstrate should clearly be described in your manuscript.

We thank the Reviewer for this comment and the opportunity to clarify the study aims and methods. Introduction section has now been completely revised and significantly shortened. The introduction section now includes a very short general overview on mpox, we removed discussion on clinical aspects of infection and diagnostic tools as not in line with the aim of this study. We included a more punctual and focused description on information relevant to this specific study aims. Moreover, the manuscript now includes broad description at the end of the Introduction section on the study aims. The primary objective of this study was to compare the characteristics of individuals who received MVA-BN vaccination with those diagnosed with mpox infection. This comparison aimed to assess the efficiency of MVA-BN vaccination delivery during the 2022 mpox vaccination campaign, specifically in terms of coverage among people at substantial risk of mpox infection and its reach to all key-populations, and in order to identify missed opportunities for prevention, having as reference for this evaluation the characteristics of people diagnosed with mpox infection. Secondarily, mpox symptoms presentation, and clinical course of disease were described to grant further insight on the underlying characteristics which might explain community transmission of the infection. Lastly, current mpox vaccination guidelines and risk factors considered in the recommendations for vaccination were compared with the demographical data, medical history and sexual behaviours of people who had been diagnosed with mpox, to facilitate a reflection on the optimal vaccine prioritization during current or future mpox outbreaks, ensuring that those at the highest risk are appropriately targeted. We indeed compared a group of people who acquired unintentionally mpox infection with a group of people who voluntarily received mpox vaccination, guided by local and international recommendations for MVA-BN mpox vaccination. These enclose risk factors and key population groups for which the vaccine was highly recommended. By doing so we took as reference the characteristics of people whore received mpox diagnosis and compared them with those of people who accesses MVA-BN vaccination. This could show whether the vaccine was correctly received by people at substantial risk of mpox infection, if there are any missed opportunities for prevention, which key populations were indeed reached and, by considering the characteristics of mpox infected people, if there are other groups or key factors which were not included in guidelines which could also be used to guide current or future vaccination policies.

If you want to write a summary of the 2022 mpox outbreak, then a lot of the elements of your current draft can be re-used. But that would be another article type, more of a review, less of a research article.

We thank the Reviewer for this important comment and apologize for the redundancy of the Introduction section. Introduction section has now been completely revised and significantly shortened. The introduction section now includes a very short general overview on mpox, we removed discussion on clinical aspects of infection and diagnostic tools as not in line with the aim of this study. We included a more punctual and focused description on information relevant to this specific study: for instance the observed sexual behaviours of people diagnosed with mpox in the current outbreak, the current international and local indications of MVA-BN vaccinations, the ways of sexual transmission of mpox observed in 2022 and the key-populations affected by the current outbreak. We believe that the Introduction is now more in line with the study aim and more useful for the correct interpretation of the study findings and the discussion section.

In case you want to keep it as a research article, there needs to be more clarity why the comparison of vaccinated versus infected individuals is appropriate, what you want to demonstrate with it, and how you selected and interpreted the parameters where you saw differences between the groups. Also an interpretation what those differences mean in the context of your research question should be provided.

We thank the Reviewer for this comment and the opportunity to clarify the study aims and methods. Introduction section has now been completely revised and significantly shortened. The introduction section now includes a very short general overview on mpox, we removed discussion on clinical aspects of infection and diagnostic tools as not in line with the aim of this study. We included a more punctual and focused description on information relevant to this specific study aims. Moreover, the manuscript now includes broad description at the end of the Introduction section on the study aims. The primary objective of this study was to compare the characteristics of individuals who received MVA-BN vaccination with those diagnosed with mpox infection. This comparison aimed to assess the efficiency of MVA-BN vaccination delivery during the 2022 mpox vaccination campaign, specifically in terms of coverage among people at substantial risk of mpox infection and its reach to all key-populations, and in order to identify missed opportunities for prevention, having as reference for this evaluation the characteristics of people diagnosed with mpox infection. Secondarily, mpox symptoms presentation, and clinical course of disease were described to grant further insight on the underlying characteristics which might explain community transmission of the infection. Lastly, current mpox vaccination guidelines and risk factors considered in the recommendations for vaccination were compared with the demographical data, medical history and sexual behaviours of people who had been diagnosed with mpox, to facilitate a reflection on the optimal vaccine prioritization during current or future mpox outbreaks, ensuring that those at the highest risk are appropriately targeted. The manuscript Abstract has also been revised accordingly. The manuscript Discussion now embraces further discussion on the implications of our findings in respect to available evidence and literature and is focused on the interpretation of our results. 

Round 2

Reviewer 1 Report

The new version of the manuscript titled “Mpox Outbreak 2022: Comparative Analysis of the Individuals Characteristics Receiving MVA BN Vaccination and People Diagnosed with Mpox Infection in Milan, Italy” has improved significantly relative to the previous one. Below are the comments for the sections that I discussed in detail in my previous review.

Abstract
The modified abstract gives a good overview of the background information, the results and the conclusion of the study. Importantly, this information is now presented in a way that stimulates the interest of the reader more.

Introduction
The authors made significant changes to the this section, as suggested. As a result, the introduction has a better structure, is more concise and summarises the important background information that is relevant to the study very well. In addition, they have now properly cited the information described in this section. I believe that the introduction does not require any further changes, other than the suggested modification below:
•Line 66: Change with the highest number of cases…” to “having the highest number
of cases…”

Methods
The authors made the recommended changes to the methods section and importantly they
included the threshold values for each HIV RNA category both here and underneath table 1.
This section is now fine as it is. The only modification needed is this one:
•Line 272: Change “e. train ad gas stations…” to “e. train and gas stations…”

Results
The authors made the modifications that I suggested in my previous review. They added the missing information median CD4 percentages to Table 1 and they modified the description of the HIV RNA data, making the description clearer to the reader.

Reviewer 2 Report

Thank you for doing major rework on your paper. It now makes much clearer, what the purpose of the paper actually is, and points nicely towards the public health importance of identifying risk groups, to ensure they can be vaccinated, without stigmatizing people by sexual orientation. 

Although I am still not 100% sure if the statistical analysis of comparing the two populations is fully relevant to support your conclusion, I would now definitely agree to have this paper published as an important piece of public health research, supporting wise use of vaccination policies.